# Mountain waves can impact wind power generation

Caroline Draxl[1], Rochelle P. Worsnop[2,5], Geng Xia[1], Yelena Pichugina[2,5], Duli Chand[3], Julie K. Lundquist[1,4], Justin Sharp[6], Garrett Wedam[7,8], James M. Wilczak[5], Larry K. Berg[3]

[1]National Renewable Energy Laboratory, Golden, CO 80401 USA

[2]Cooperative Institute for Research in the Environmental Sciences, University of Colorado Boulder, Boulder, CO 80309 USA

[3]Pacific Northwest National Laboratory, Richland, WA 99352 USA

[4]Department of Atmospheric and Oceanic Sciences, University of Colorado Boulder, Boulder, CO 80302 USA

[5]National Oceanic and Atmospheric Administration/Earth System Research Laboratory, Boulder, CO 80305 USA

[6]Sharply Focused, LLC, Portland, OR 97232 USA

[7]Avangrid Renewables, Portland, OR 97209 USA

[8]Natural Power, Seattle, 98121, USA

*Correspondence to:* Caroline Draxl (caroline.draxl@nrel.gov)

**Abstract.** Mountains can modify the weather downstream of the terrain. In particular, when stably stratified air ascends a mountain barrier, buoyancy perturbations develop. These perturbations can trigger mountain waves downstream of the mountains that can reach deep into the atmospheric boundary layer where wind turbines operate. Several such cases of
mountain waves occurred during the Second Wind Forecast Improvement Project (WFIP2) in the Columbia Basin in the lee of the Cascade Mountains bounding the states of Washington and Oregon in the Pacific Northwest of the United States. Signals from the mountain waves appear in boundary-layer sodar and lidar observations as well as in nacelle wind speeds and power observations from wind plants. Weather Research and Forecasting (WRF) model simulations also produce mountain waves and are compared to satellite observations, lidar, and sodar observations. Simulated mountain wave wavelengths and wave
propagation speeds (group velocities) are analysed using the Fast Fourier Transform. We found that not all mountain waves exhibit the same speed and conclude that the speed of propagation, magnitudes of wind speeds, or wavelengths are important parameters for forecasters to recognize the risk for mountain waves and associated large drops or surges in power. When analysing wind farm power output and nacelle wind speeds, we found that even small oscillations in wind speed caused by mountain waves can induce oscillations between full rated power of a wind farm and half of the power output, depending on
the position of the mountain wave's crests and troughs. For the wind plant analysed in this paper, mountain wave induced fluctuations translate to approximately 11% of the total wind farm output being influenced by mountain waves. Oscillations in measured wind speeds agree well with WRF simulations in timing and magnitude. We conclude that mountain waves can

impact wind turbine and wind farm power output and, therefore, should be considered in complex terrain when designing, building, and forecasting for wind farms.

## 1 Introduction

As wind farm deployment in the United States and worldwide continues to increase, contributions from renewable wind energy production to the electrical-generation portfolio are also increasing (AWEA Data Services 2017; Global Wind Energy Council 2018). The U.S. Department of Energy's (DOE's) Wind Vision study (DOE 2015) mapped out a target scenario for wind energy to provide 35% of the United States' electricity demands by 2050. Wind plants are already deployed in areas of complex terrain, and will continue so, to satisfy that portfolio. Complex terrain, herein defined as terrain with irregular topography (e.g., mountains, valleys, coastlines, and canyons), can modify the flow within and far downstream of the terrain.

One area of complex terrain where numerous wind farms are deployed is the Columbia River basin in the Northwest United States, which is located east of the Cascade Range. The Cascade Range extends from southern British Columbia through Washington and Oregon to Northern California, for 1,100 km (Wikipedia), with a width of 130 km. Volcanic summits in the area reach up to approximately 4,000 m above mean sea level. During westerly flow, the Cascade Range poses an obstacle that impacts the weather and modifies the wind flow to the east of the Cascade Range, which impacts wind farm production of the deployed wind power plants in the area (Figure 1).

During westerly winds with stable atmospheric conditions, air ascends the Cascade Mountains and strong buoyancy perturbations can develop in the form of mountain waves, or lee waves, downstream of the Cascade Range. The area is prone to these conditions primarily during the cold and transition seasons, mostly during spring. Mountain waves may be nearly stationary, propagating downwind, vertically propagating, or trapped. Vertically propagating waves are relevant to wind energy to the extent that they can lead to downslope windstorms. Trapped lee waves are relevant to wind energy because they occur in the lowest 1–5 km of the troposphere (AMS glossary, Durran 1990). With their horizontal wavelengths between 5 km and 35 km, trapped lee waves have anecdotally been recognized to impact wind farm production in the area, particularly if stationary.

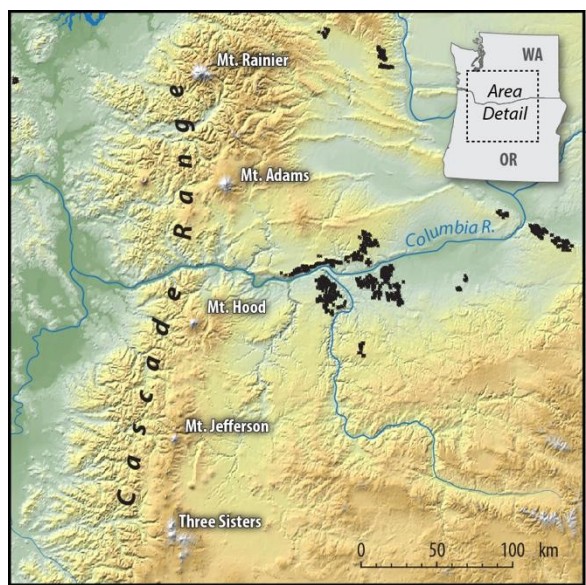

**Figure 1. Map with the location of the major volcanoes (white labels) and wind farms (black dots) in the area. Figure by Billy Roberts, NREL.**

The mountains of the Cascade Range can also block atmospheric flow and create a wake behind them. Mountain wakes are usually accompanied by significant drag and deceleration of low-level flow (Wells et al., 2008). During westerly winds, such mountain wakes are relevant for wind energy in the Columbia River Basin as they create meandering bands of low wind speeds that can extend hundreds of kilometres downwind and decrease power output from wind farms. Their exact timing and location are hard to predict as they meander.

Mountain waves and wakes can and commonly do occur concurrently within the Columbia River basin (Wilczak et al. 2019, Pichugina et al. 2020). In this region, mountain wakes mostly occur downstream from Mt. Hood and Mt. Adams (Figure 1), as seen from satellite observations and model simulations (examples are shown in Figure 4 and Figure 7).  Both mountain waves and wakes impact wind plants and their power output. Because these phenomena can occur simultaneously and under the same conditions, distinguishing their relative impacts can be challenging. Taking advantage of the rich data set of the

Second Wind Forecast Improvement Project (WFIP2; Shaw et al. 2019 and Wilczak et al. 2019) and mesoscale simulations from the Weather Research and Forecasting (WRF) model, this paper focuses on one phenomenon only—the impact of mountain waves on wind farms.  Many studies have analysed mountain wakes for the last decades (e.g., Lindsay 1962; Bourgeault et al. 1993; Klemp and Lilly 1997; Doyle and Durran 2002; Durran 2003; Smith 2003; Smith and Broad 2003; Grubišić and Billings 2007; Smith et al. 2007; Mahalov et al. 2011; Nappo 2012; Vosper et al. 2012; Miglietta et al. 2013;

Durran 2015; and Fritts 2015). However, even though one article (Rasheed et al. 2014) mentions that mountain waves result in horizontal and vertical wind shear, which can significantly impact wind power production, none quantified that impact. Therefore, it is our goal to document for the wind energy community, in particular for forecasters and the wind energy industry, the importance of considering mountain waves in operations and wind plant deployment. A second goal of this paper is to

what degree the mesoscale WRF model is able to capture mountain wave characteristics in the complex terrain of the Columbia Basin, a key region for wind energy production.

This paper is structured as follows: In the next section, we describe the measurements and model simulations used. We then identify and quantify mountain waves in Section 3 from a meteorological perspective and as simulated by WRF, before we analyse the impact of mountain waves on wind farm output using nacelle winds and supervisory control and data acquisition (SCADA) data. In Section 4, we provide a discussion relating our findings to practical aspects of mountain waves in forecasting and operations then conclude in Section 5 with recommendations for actions during mountain wave events.

## 2 Data and Methods

Our analysis is based on the extensive measurement network from the WFIP2 (Shaw et al. 2019 and Wilczak et al. 2019). WFIP2 is a DOE and National Oceanic and Atmospheric Administration (NOAA)-funded program aimed at improving the accuracy of numerical weather prediction (NWP) model forecasts of wind speed in complex terrain for wind energy applications (Wilczak et al. 2019; Banta et al. 2019; Bianco et al. 2019; Olson et al. 2019; Pichugina et al. 2019, 2020). Measurements were collected during an 18-month field campaign between October 2015 and March 2017 in the Columbia River Gorge and the Columbia Basin (Figure 1). For this paper, measurements from remote sensing instruments are used (Section 2.1) to identify mountain waves through time-series analysis, spectra, and statistics. Satellite images (Section 2.2) further help identify mountain waves, and WRF model simulations (Section 2.3) support our analysis. Nacelle wind speeds and power output from a wind farm in the area portray the influence of mountain waves on wind plants.

### 2.1 WFIP2 observations

To analyse wind flow variability during mountain wave events, we use profile measurements from lidars and sodars (Sections 2.1.1 and 2.1.2) that were deployed in the WFIP2 research area. These instruments continuously operated during the 18-month experiment providing real-time data. These quality-controlled data are openly available to the public through the Data Archive and Portal (DAP; https://a2e.energy.gov/data). Proprietary nacelle wind speeds from a wind farm in the area, as well as its power output (Sections 2.1.3), quantify the impact of mountain waves on wind farms.

### 2.1.1 Lidar data

Several profiling and scanning lidars were deployed as part of the WFIP2 field campaign. This study uses measurements from the scanning Doppler lidar and the wind profiling lidar at the Wasco site (Figure 2). The WindCube profiling lidars sample line-of-sight velocities sequentially in four cardinal directions along a nominally 28° azimuth from vertical and a nominal temporal resolution of 1 Hz (Aitken et al. 2012; Rhodes and Lundquist, 2013), simultaneously sampling ten range gates centered on 40, 60, 80, 100, 120, 140, 160, 180, 200, and 220 m AGL. These lidars provide estimates of wind speed, wind

direction, and vertical velocity within the surface layer and boundary layer up to 250 m AGL (Bodini et al, 2019) that were also used for the selection of mountain wave cases. Basic quality control, requiring that an individual line-of-sight (LOS) velocity be measured with a carrier-to-noise ratio greater than -22 dB, has been applied to these data. The two-minute averages are based only on the 1-Hz LOS with CNR exceeding -22 dB. Lidars require a sufficient number of scatterers for a return signal, so clean air conditions have lower availability (Aitken et al. 2012).

### 2.1.2 Sodar data

In this paper we use sodar measurements from the Wasco and Van Gilder Road sites. At Wasco, the ART VT-1 sodar model was deployed, which is a monostatic phased-array Doppler sonic detection and ranging (sodar) system. It provides a "virtual tower" for obtaining remote measurements of the wind profile up to a height of approximately 300 m at a vertical resolution of 10 m. The system includes a 48-element acoustical array. At Van Gilder Road, which is close to Wasco, a triton wind profiler is used. This profiler measures wind speed, direction, and turbulence intensity at heights from 30 m to 200 m above ground every 10 min. The quality-controlled sodar data are stored on the DAP. Information about setup and filtering for the Wasco and Van Gilder Road sodars can be found in Atmosphere to Electrons, 2020 (a) and Atmosphere to Electrons, 2020 (b), respectively.

### 2.1.3 Nacelle winds and turbine power output

We use data from approximately 100 wind turbines from a wind farm in the WFIP2 region to assess how mountain waves influence observed wind speed and power output. The wind farm is located north of the Columbia River Gorge and experienced mountain wave events during the WFIP2 field campaign. From the turbine nacelles, we use 80 m, 10 min averaged wind speed data and 10 min averaged power output. Data from a single turbine, as well as spatially aggregated winds across an entire wind plant, are compared with outputs from corresponding WRF simulations (see Section 3.2).

### 2.2 Satellite images

The mountain waves are detected using visible clouds features from the satellite observations. We utilized both polar orbiting and geostationary satellite observations to locate the mountain waves downwind of mountain peaks. The cloud features are retrieved from the Geostationary Operational Environmental Satellite (GOES-14) routine observation over the continental United States. To have the best cloud contrast and spatial separation, we used 1 km resolution pixels from band 1 (approximately 630 nm) of the GOES-14 satellite. The GOES retrieved mountain wave features in the form of clouds are compared with Moderate Resolution Imaging Spectroradiometer (MODIS) satellite observations for better understanding. The comparison in both satellites looks reasonable, though the MODIS observations show finer cloud features due to higher spatial resolution (0.25 km). Since MODIS observations have higher spatial resolution but limited temporal resolution (only one

MODIS satellite per day passes over the Columbia Basin), considering the temporal evaluation of waves, we decided to use the GOES-14 observations at a temporal resolution of 30 min.

## 2.3 WRF simulations

Model simulations at 5 min resolution produced with the WRF model version 3.7.1 augment the observational analysis. We use model output from an inner domain with a 750 m grid spacing, that was nested within a larger domain at 3 km grid spacing (Figure 2). ERA-Interim reanalysis data (Dee et al. 2011) provide initial and boundary conditions. We used the Mellor-Yamada Nakanishi and Niino Level 2.5 boundary layer and surface layer schemes (Nakanishi and Niino 2009) as they were improved upon within WFIP2, the Morrison double-moment microphysics scheme, the Rapid Radiative Transfer Model for Global

Circulation Models, simple diffusion, and vertical velocity damping (Skamarock et al. 2008). This model setup has been successfully used in DOE's Mesoscale-to-Microscale Coupling project and was constructed with input from modeling experts in the project (e.g., Haupt et al. 2017).

Computations were carried out using 88 vertical levels, up to 10,000 hPa, which were spaced approximately 5 m apart in the lowest 20 m, with the grid spacing increasing continuously beyond that. This allows for a vertical resolution of 8–10 m within

150 the turbine rotor layer (approximately 20–150 m above ground level (AGL)).

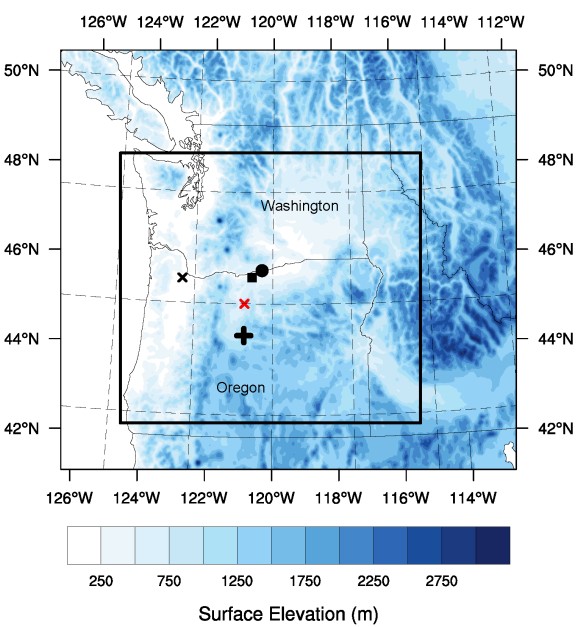

**Figure 2. WRF modelling domains. The rectangle denotes the area of the 750 m domain. U.S. state boundaries are indicated. The black x denotes the location of Troutdale, the cross Prineville, the square Wasco and Van Gilder Road, the circle the location of**
155 **the wind farm in the area, and the red x the location in the WRF domain where profiles are plotted in Figure 5. The Columbia River cuts through the Cascade Range at the border between the state of Oregon to the south and Washington to the north.**

## 3 Results

Each week throughout the WFIP2 field program, scientists and wind energy forecasters reviewed the daily weather in the region and wrote a brief synopsis in an event log (Wilzcak et al. 2019), assessing the significance of the key phenomena (Pichugina et al. 2020) that impacted wind power generation. Key phenomena were categorized by having a "High", "Medium", and "Low" importance (%) for wind energy (WE), which was estimated by analysing all available observations, Bonneville Power Administration (BPA) power generation and schedule error, power ramps, and the performance of the NOAA's High-Resolution Rapid Refresh model (Figure 3a). From this event log, we found 95 days during the 18 months (548 days) where mountain wave activity was indicated by a meteorologist, so that mountain waves were present at least 17% of the time. Each of the key phenomena were categorized further by three levels of significance (Potential, Interesting/Relevant, or Not currently of interest). The significance of mountain wave cases for each level of importance relative to all phenomena is shown in Figure 3a.

As noted in the introduction, topographic wakes often occur simultaneously with mountain waves. During WFIP2, topographic wakes were recorded in the event log 15% of the time, based on analysis of wind speed observations, comparisons of observations in waked versus nonwaked areas, and their appearance in satellite images and horizontal slices of simulated wind speeds. Distributions of mountain waves and topographic wakes (Figure 3b) show a high frequency of both events during spring months.

Scanning this event log as well as lidar and sodar observations, we identified 2 days where mountain waves had a strong presence over the area and impacted wind farms in the Columbia Basin. The first day (11 November 2016) is documented in Wilczak et al. (2019). In this paper, we focus on the second day (24 September 2016) because of the availability of measurements and SCADA data and the presence of typical characteristics of mountain waves.

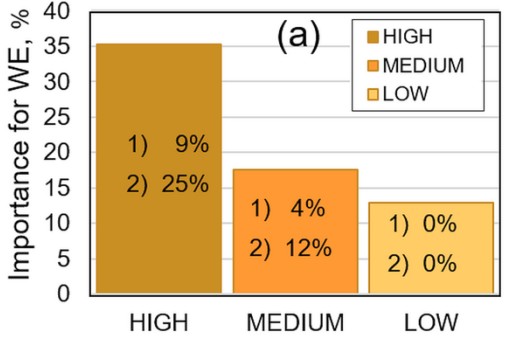 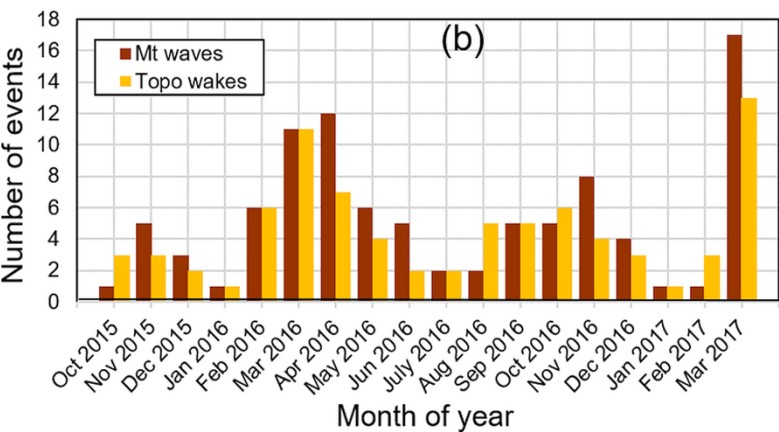

**Figure 3: (a) Distribution of days during WFIP2 that exhibited observed phenomena that were ranked as of HIGH, MEDIUM, and LOW importance to wind energy according to the Event Log. The frequency of cases per level of importance that are considered a potential (1) or interesting/relevant (2) case for wind energy is given in each bar. (b) Distribution of mountain waves and topographic wakes observed during WFIP2 according to the Event Log.**

### 3.1 Analysis of model simulations on 24 September 2016

Reichmann (1978) and Mastaler and Renno (2003) state that the best conditions for mountain waves are: (i) the presence of a stable air mass, (ii) wind speeds aloft must be greater than about 8 m s$^{-1}$ at ridge level, (iii) the wind direction should be nearly constant throughout the stable layer, (iv) the wind speed should be constant or increasing with altitude, and (v) the wind direction should be within 30 degrees of normal to the perturbing ridge. From these conditions it is deduced that the Scorer parameter (Scorer 1949), an indicator for mountain wave development, should decrease with altitude. On 24 September 2016, all the above conditions were met in the model simulations, as will be discussed in this section.

On 24 September 2016, stationary mountain waves east of the Cascade Mountains were caused by flow from north-westerly directions. That flow, in turn, was forced by a low-pressure system over the western half of the continental United States (see Appendix). The area of the Columbia Basin was covered by clouds at 00:00 UTC, which completely dissolved by 07:30 UTC (not shown).

Mountain waves can be seen in the simulated horizontal wind field at 100 m AGL (Figure 4 (a)) as relatively thin and similarly spaced oscillating strips of high and low wind speeds oriented approximately perpendicular to the wind direction. They are triggered by the flow over the Cascade Mountains already around 23 September 2016, 16:00 UTC, and between 20:00 and 22:00 UTC they take over the whole area, impacting Prineville and a wind farm in the WFIP2 region. An elongated wake extending downstream from Mt. Hood (large triangle in Figure 4 (a)) is narrowed to a meandering band when the wakes cover the area. Wakes are also discernible downstream from Mt. Jefferson, Sisters, and Broken Top.

A cross section of the simulated horizontal wind field, potential temperature, and the PBL top from west to east on 24 September 2016, at 04:00 UTC, centered over a wind farm (Figure 4 (b)), further shows the appearance of mountain waves in the simulations, up to approximately 4.5 km above sea level, as do oscillating patterns in the vertical wind speeds (Figure 4 (c)).

The stratification of the atmosphere during the occurrence of mountain waves is shown for Troutdale, a location west (and therefore upstream) of the Cascade Mountains, and for a location in the centre of the area where waves occur (Figure 2 and Figure 5). The atmosphere is stably stratified, except from 00:00–03:00 UTC where a well-mixed layer exists below the approximately 1500 m crest height up to approximately 1 km. The simulated wind speed profiles show winds between approximately 6 m s$^{-1}$ and 10 m s$^{-1}$ up to approximately 2.5 km, decreasing, with a high wind shear, above 2 km. In the centre of the domain, during the time where mountain waves are present, the wind speeds are higher than at Troutdale but exhibit a similar wind shear above approximately 3 km. The stable stratification, wind speed magnitudes (satisfying the constraint from Mastaler and Renno (2003) that wind speeds aloft must be greater than about 8 m s$^{-1}$ at ridge level), constant wind speeds below 2 km, and increased wind speeds above that, are favourable conditions for the development of mountain waves.

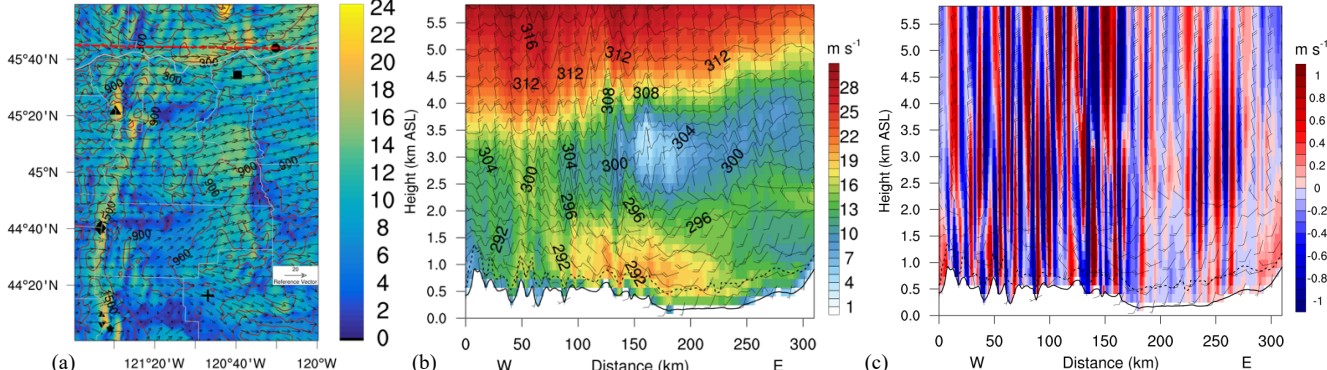

**Figure 4. (a) Simulated wind field at 100 m AGL. The big triangle, diamond, cross, square and circle denote locations of Mt. Hood, Mt. Jefferson, Prineville, Wasco instrument site, and the wind farm, respectively. Sisters and Broken Top mountains located in lower left corner are shown by smaller triangles and a star, respectively. Contours show terrain elevation every 300 m. The red dashed line indicates the transect along which cross sections in (b) and (c) are taken. Note that the cross sections in (b) and (c) extend beyond the bounds of this plot. (b) Cross section of horizontal wind speed from west to east through the wind farm. Wind speeds are colour shaded, lines denote potential temperature [K], the boundary layer top is dashed. (c) Cross section of vertical wind speeds through the wind farm at 04:00 UTC on 24 September 2016.**

Mountain waves are possible when the nondimensional mountain height is in the order of 1 (Mastaler and Reno 2003). Therefore, we calculated the nondimensional mountain height and the Scorer parameter (Figure 6) from the WRF simulations at Troutdale. The nondimensional mountain height was calculated using the bulk method by Reinecke and Durran (2007), using an average free stream velocity between 1,500 and 7,000 m, a mountain height of 1,500 m, and a Brunt–Väisälä frequency between 0 m and 7,000 m. Mountain waves are possible because the nondimensional mountain height was high during the entire day (Figure 6). In fact, the horizontal wind field shows waves until approximately 15:00 UTC.

The Scorer parameter (Eq. 1) is a further measure to determine whether mountain waves develop:

$$l^2 = \frac{N^2}{U^2} - \frac{1}{U}\frac{d^2U}{dz^2} \tag{1}$$

where $U(z)$ is the speed of the basic-state flow and $N(z)$ is the Brunt-Väisälä frequency, with $z$ being the vertical coordinate (Durran, 2003). According to Scorer (1949), waves are possible if atmospheric stability decreases or wind speed increases with height (Lindsay 1962). In our case, wind speeds increase with height (Figure 5). Moreover, when the Scorer parameter is nearly constant with height, conditions are favourable for vertically propagating mountain waves. Trapped lee waves can be expected when the Scorer parameter $l$ decreases with height. Figure 6 (b) shows that the Scorer parameter from model output at Troutdale at 01:00 UTC increases with height up to ~200 m and is mostly constant above that. At 04:00 UTC, it also increases with height up to ~200 m, exhibits multiple maxima until about 1100 m, and is nearly constant with height above that. Multiple maxima indicate that multiple wave systems may occur simultaneously. The slight change of the profile in time indicates that the simulated mountain waves may change their propagation characteristics slightly over time.

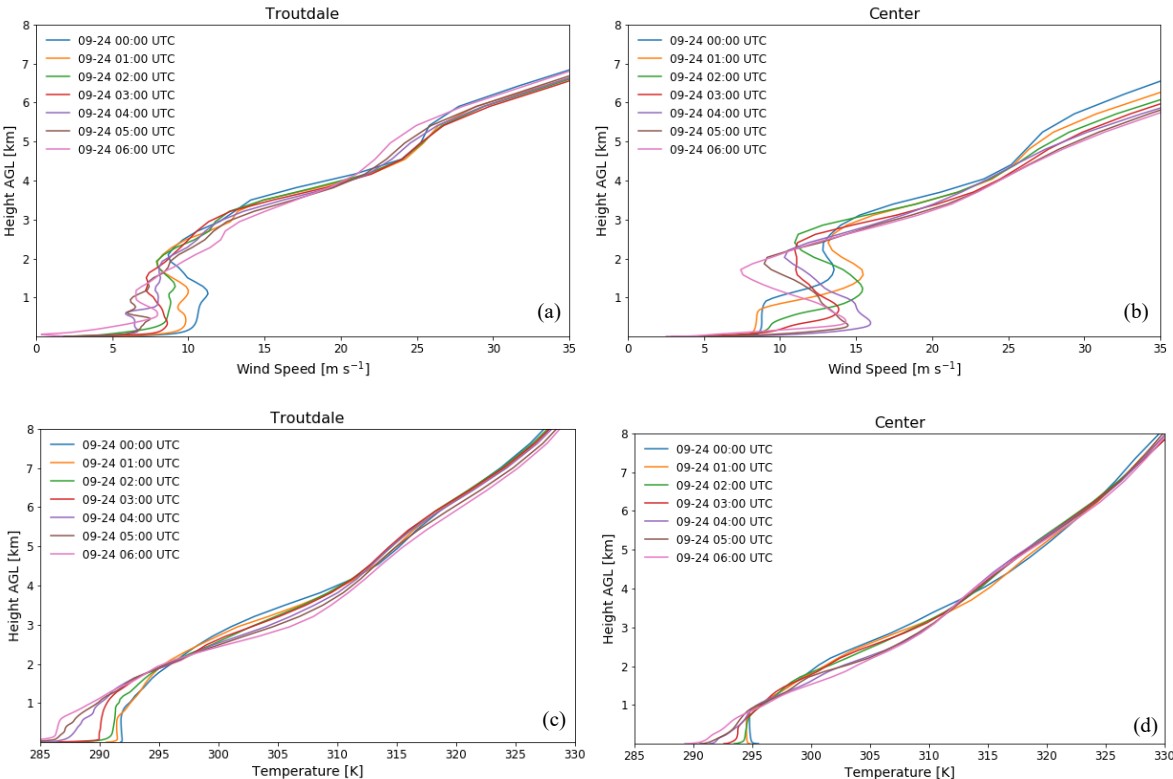

**Figure 5. Simulated WRF wind speed [(a), (b)] and potential temperature profiles [(c), (d)] at Troutdale [(a), (c)] and a location in the centre of the modelling domain, east of the Cascades, [(b), (d)] during mountain wave activities.**

## 3.2 Comparison of model simulations with observations on 24 September 2016

We will now compare the model simulations with observations to see how well the model captures mountain waves. We will look at satellite images and lidar and sodar observations at Wasco (Figure 2). Note that lidar and sodar observations represent data collected at a single point in space; therefore, signals in these data will indicate nonstationary mountain waves.

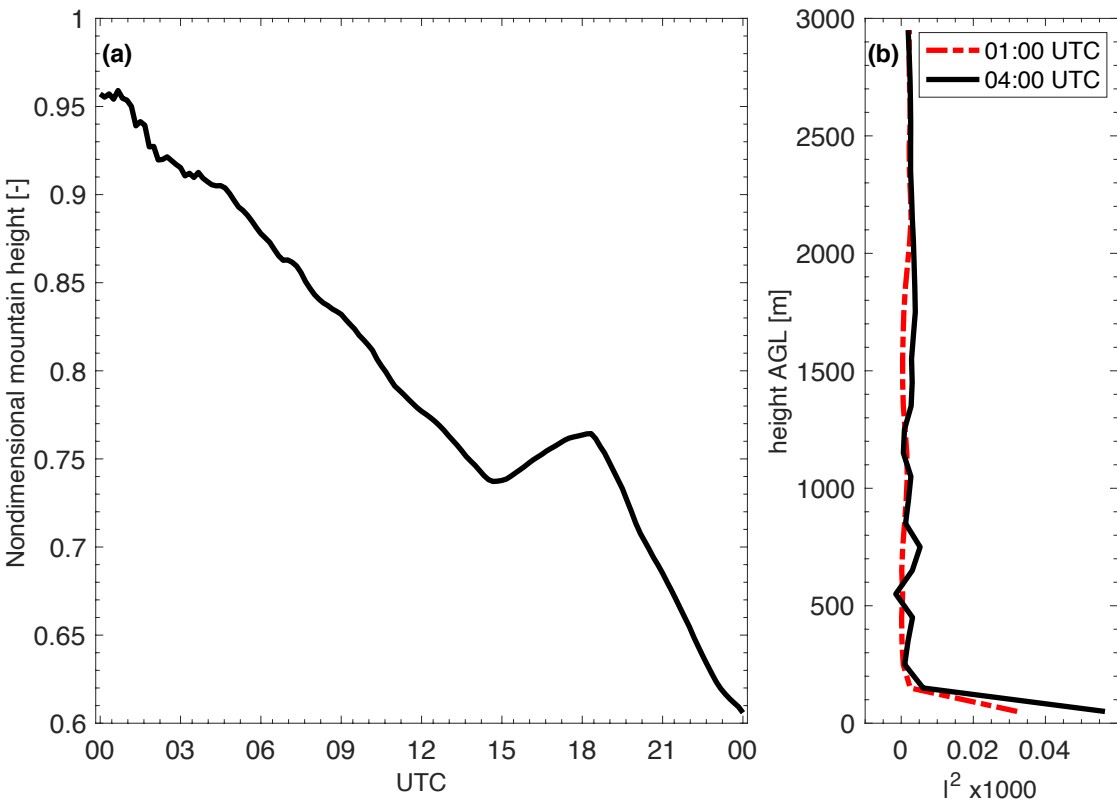

**Figure 6. (a) Nondimensional mountain height as a function of time of day (UTC), (b) Scorer parameter at 01:00 UTC (dashed red), and at 04:00 UTC (solid) from model simulations.**

### 3.2.1 GOES satellite imagery

GOES visible reflectances at 1000 m resolution and 630 nm wavelengths (Figure 7 (a)) show a wavy cloud pattern similar to
the simulated wind field in Figure 4 (a) on 23 September 2016, from 19:30 UTC until sunset (shown for 22:00 UTC on 23 September 2016. After sunset, visible satellite images cannot reveal any signals. The appearance of mountain waves during that time matches with the model simulations in which mountain waves appeared starting on 23 September 2016, around 16:00 UTC.  From the clouds, a wavelength of approximately 8 km was deduced (Figure 7 (a, b)). The wavelength is calculated by averaging the cloud reflectance from 44.4°N to 46.6°N along 121°W to 121.3°W, shown by the meridional distribution in
Figure 7 (b). Note the spatial heterogeneity in the cloud field (Figure 7 (a)), which indicates similar variability in the manifestation of mountain waves.

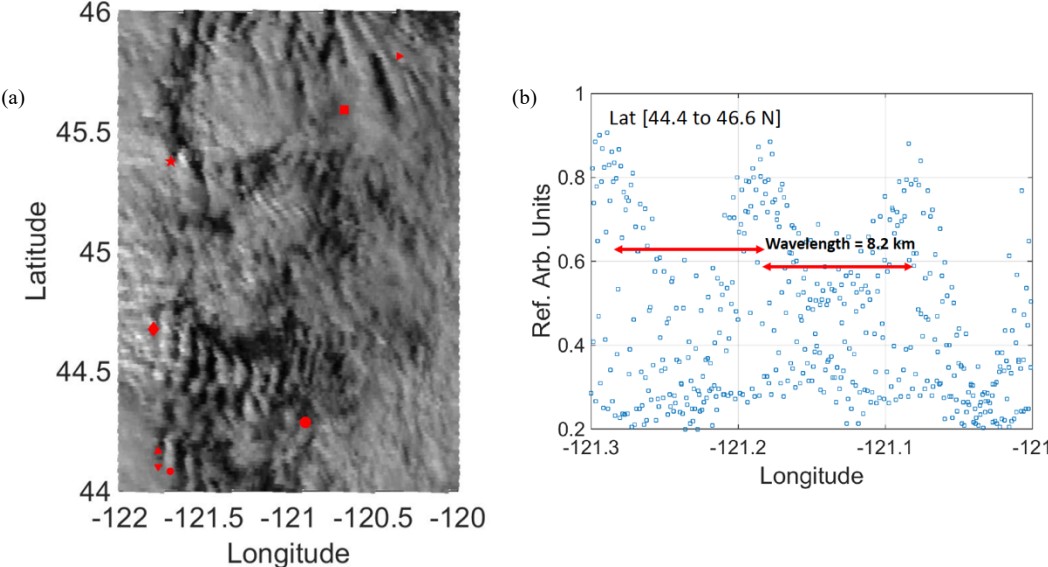

**Figure 7. (a) GOES-14 satellite image on 23 September 2016, at 22:00 UTC. The red dots represent the same locations as in Figure 4. (b) The dots denote cloud reflectance in arbitrary units (counts or normalized data) covering latitude 44.4 to 46.6 N at 22:00 UTC.**

### 3.2.2 Lidar and sodar observations

Observations at fixed locations (such as from lidar or sodar) can reveal the presence of trapped lee waves through temporal fluctuations in the lee-wave pattern (Bougeault et al. 1993, Wilczak et al. 2019). Periods of alternating high and low wind speeds were observed at Wasco from all collocated remote sensing instruments as well as in the simulated horizontal wind field (Figure 8). Good agreement is found between data from all instruments (Figure 8a–c), as waves manifest in all instruments starting near 02:00 UTC, increasing in amplitude until a maximum near 10:00 UTC, then decreasing. Clear patterns of waves are discernible in both measured and simulated wind speeds. However, during the high-wind-speed periods, WRF overpredicted the wind speed from 06:00 to 09:00 UTC. Further, the phase of the waves in WRF did not always match that of the observations. In general, the waves are well captured in time and magnitude.

### 3.2.3 Wavelengths and speed of wave propagation observed on 24 September 2016

From an operational forecasting perspective, knowing when mountain waves will influence wind power and for which period of time can be valuable for forecasting for power trading, and forecasting the balancing requirements in the power system – from both a regulatory and economic perspective. Nearly stationary mountain waves lend to less short-term volatility in energy production, but potentially a still large deviation from scheduled production and therefore large balancing requirements for the duration of the event. Such stationary events can lead to large and costly imbalances for power producers. Large wavelengths

(>18 km) can exacerbate the imbalance for both grid operators and power producers by reducing (or enhancing) production over multiple wind farms at once, whereas shorter wavelengths (where several wavelengths occur within one wind production region) will tend to have some areas of enhanced production and other areas of reduced production, resulting in a beneficial

netting effect. Quickly propagating mountain waves produce the netting effect on a temporal scale, so that while short-term imbalances can be large and require costly balancing reserves, longer term imbalances may be small. Balancing costs at all described time scales are important to grid operators and energy producers, but they require different planning. We therefore investigate whether our model simulations are able to forecast the speed of the mountain wave propagation, as well as their wavelengths.

From the spatial pattern of mountain waves in the 100 m wind speeds, we extract wind speeds along 45.6 degrees north latitude and calculate the power spectrum using the fast Fourier transform (FFT) (Figure 9). The spatial pattern of the waves at 50 and 200 m are similar (not shown). At this latitude, most of the WFIP2 sodar sites are located. Evidently, most of the power variances are explained by low-frequency waves and large wave patterns (Figure 9b). For wavelengths shorter than 8 km the associated power variance is negligible. From the analysis in Section 3.2, we identified that mountain wave wavelengths range

from 8–18 km. In that range, the bulk of the power variance occurs between 23 September 22:00 UTC, and 24 September 04:00 UTC. Therefore, we reconstruct the wind field by filtering it with respect to wavelengths between 8 km and 18 km.

To confirm our choice of wavelength range, we show Hövmöller diagrams of the original and reconstructed hub-height wind speed (Figure 10) at the targeted latitude. There is a mountain wave event particularly distinguishable between 23 September 22:00 UTC, and 24 September 04:00 UTC (Figure 10 (a) and power variance in Figure 9 (b)), which is well captured by the

reconstructed wave pattern (Figure 10 (b)). To determine the wave period, Figure 10 (c) shows the power spectrum of the reconstructed (8-18 km wavelength) and observed hub-height wind speed. We have removed the low-frequency wave signals (24 h and 12 h) from both the observed and simulated time series to focus on high-frequency waves. We chose the wave periods to be between 1 h and 4 h because that is within the time range of our interest (22:00 UTC to 04:00 UTC) and it explains the majority of the power variance by the simulated mountain waves (Figure 9 (b)).

Finally, we reconstruct the simulated wind field at each time step using band-pass filtering (FFT), wavelength and wave period constraints (Xia et al. 2020), and compare that with the observations (Figure 11). Because the mountain wave event that we are interested in is particularly visible in the simulations between 22:00 UTC and 04:00 UTC the next day, we only plot that period for comparison. The reconstructed wind speed time series shows a 1-h shift compared to the observations. In addition, we can deduce a wave period of 2.5 h; with a wavelength of 8-18 km, we estimate the wave speed to be 1.5 m s$^{-1}$. The results

seem to be sensitive to the chosen grid point and the period of interest (not shown). For instance, we performed a similar analysis using a grid point that is about 7 km (10 grid points) away from the original one. The simulated wind field looks similar to that of Figure 11 but the resemblance with the observations is weaker.

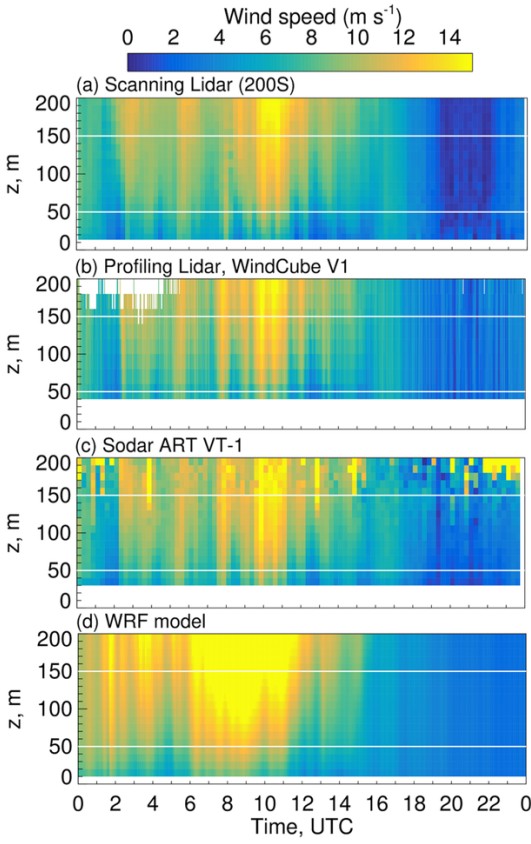


**Figure 8. Observed wind speeds up to 200 m above ground as a function of time from the (a) scanning Doppler lidar (200S), (b) the profiling lidar (Wind Cube, V1), (c) the sodar (ART VT1), and (d) simulated wind speeds at Wasco on 24 September 2016. Two horizontal white lines in each panel indicate the 50–150 m layer where turbines operate.**

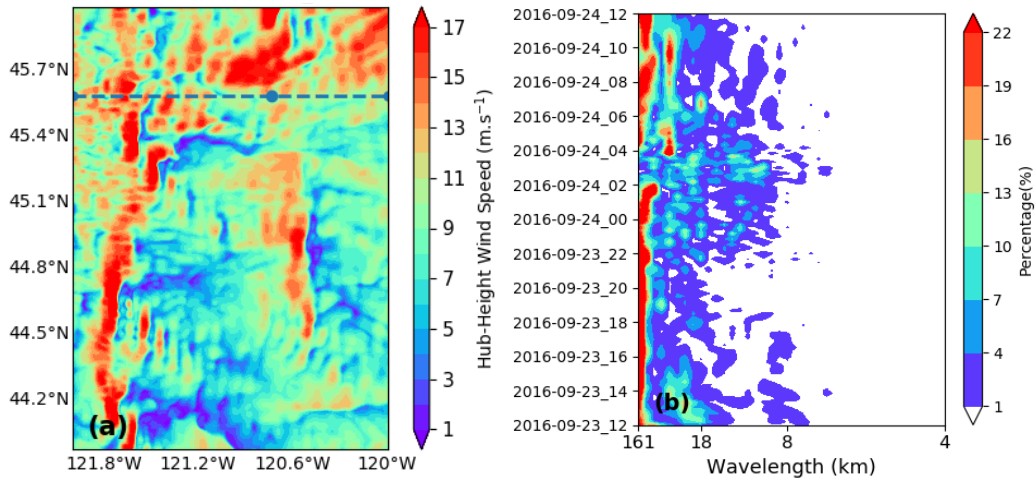


**Figure 9. (a) Simulated horizontal wind speeds [m s⁻¹] at 100 m on 24 September 2016, 02:00 UTC; the dashed line at 45.6 degrees north indicates where the FFT was taken. The dotted point represents the location of the sodar site at Van Gilder Road close to Wasco. (b) Hövmöller diagram of power variance with respect to wavelength at the targeted latitude.**

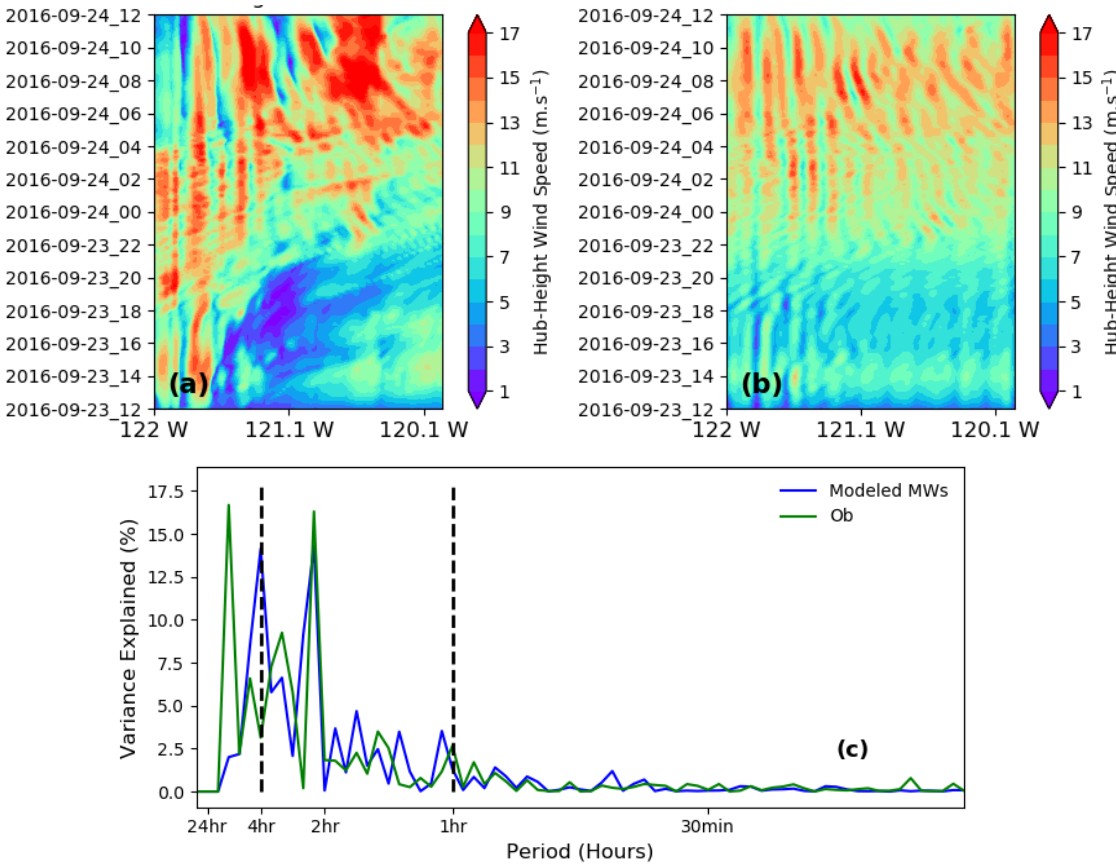


**Figure 10. (a) Hövmöller diagram of simulated hub-height wind speeds at the targeted latitude; (b) Same as (a), but showing the filtered wind speeds with wavelengths from 8-18 km; (c) Observed (green) and reconstructed (blue) power spectrum on the time domain for 24 September 2016. The simulated FFT was reconstructed with wavelengths from 8-18 km. The dashed lines indicate the wave period of interest from 1-4 hours.**


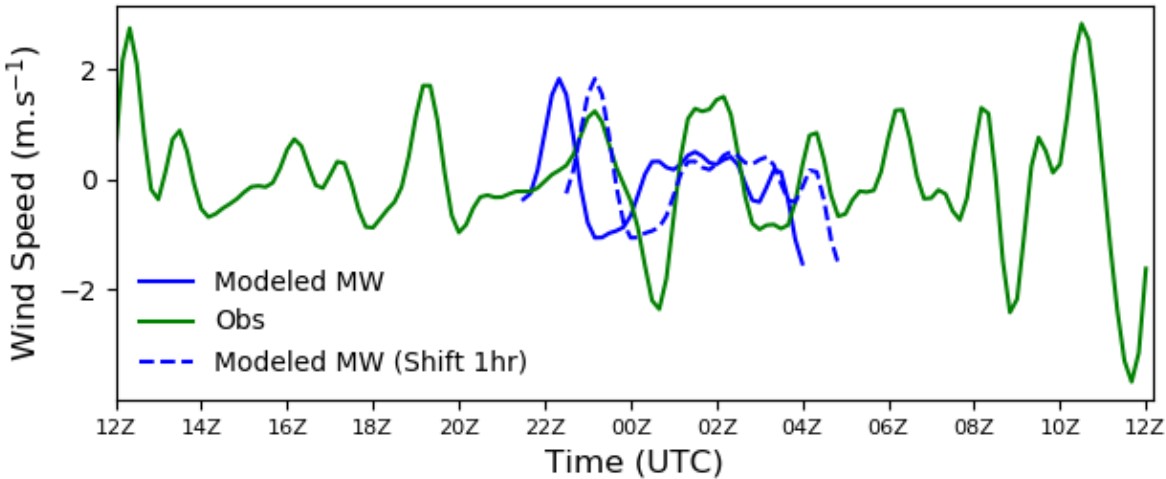

**Figure 11. 100 m (green) observed and (blue) reconstructed simulated wind speed time series from 23 September 2016, 12:00 UTC, until 24 September 2016, 12:00 UTC. The simulated wind speeds were reconstructed with wavelengths from 8-18 km, and periods from 1-4 h, while the observed winds were reconstructed with periods from 1-4 h. The simulated mountain waves were plotted from 22:00 UTC to 04:00 UTC because the mountain wave event that we are interested in is particularly visible at that time. The dashed time series indicates the simulated mountain waves that were shifted by 1 h.**

### 3.3 Impact of mountain waves on power output

The impact of mountain waves on wind power plant output in the Pacific Northwest has been anecdotally recognized by wind energy meteorologists for about a decade, and operational meteorologists know to expect additional power-generation volatility when mountain waves are present. The first time this impact was documented in a peer-reviewed journal was in Wilczak et al. (2019). Wilczak et al. (2019) confirmed signals in wind plant power output through spectra by showing that the frequency range of dominant energy is consistent with the period of mountain waves identified via satellite and wind speed observations. In this paper, we provide additional proof of the impact of mountain waves on power output by analysing wind farm power output from another wind farm in the area on a different day. We use nacelle wind speeds and model wind speeds as well as individual turbine and total farm power output.

First, power output from a wind plant in the study area is compared to measured wind speeds at the turbines and WRF output. Figure 12 shows the direct influence that mountain waves can have on power output of a single turbine. The number of wave crests (approximately 6) agrees well with the lidar and sodar observations in Figure 8. During the time when mountain waves were present (00:00–12:00 UTC), the winds were fairly strong (approximately 10 m s$^{-1}$). Oscillations in measured wind speeds were around 5 m s$^{-1}$ and agree well with WRF simulations in timing and magnitude. These oscillations in wind speed correspond with oscillations in observed turbine power. During this particular event, these oscillations are at such a critical point (region 2) in the power curve that small oscillations about the overall mean flow can make all the difference between full rated power (approximately 2.3 MW) or 1 MW of power at any given time.

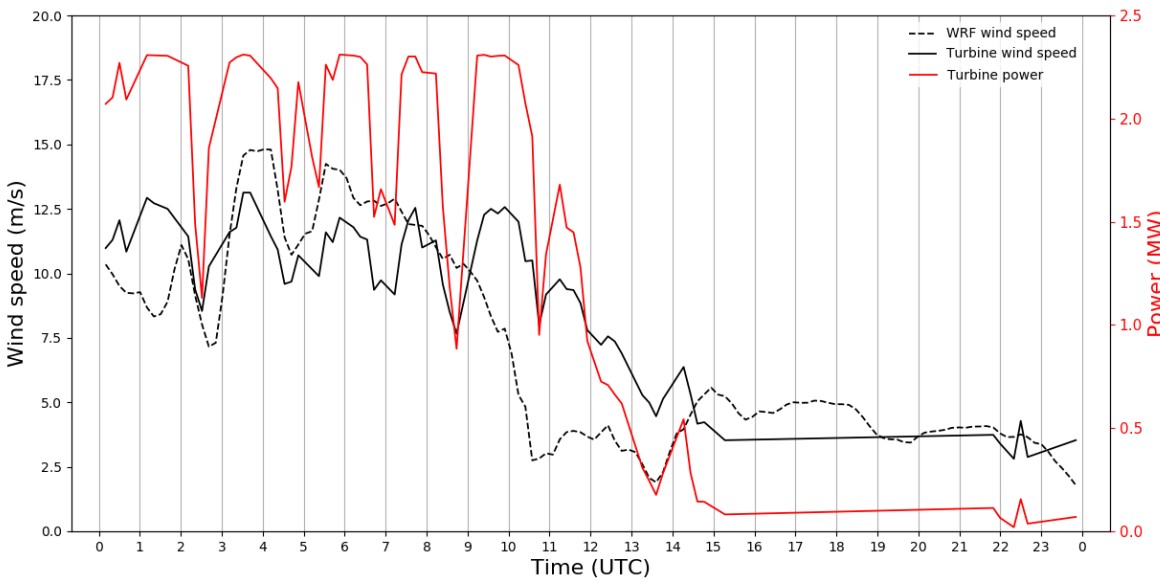


**Figure 12. Time series of simulated 80 m wind speed from WRF (dashed line), observed power output from one turbine near the middle of the wind farm (red), and wind speed measurements from that turbine (black solid line).**

Mountain waves can influence the total wind farm power output as well. The time series in Figure 13 shows oscillations in

total power output from the entire wind farm (green), and total power output from two other wind farms in the area (orange and blue). Oscillations of approximately 25 MW exist in averaged power at the wind plant (shown in Fig. 15 as percentage) and did not get cancelled out by alternating wave influences at different locations in the wind farm. Averaged wind speeds for that wind farm indicate similar oscillations (not shown). Oscillations in power output are also visible at the other two wind farms (although those oscillations are not as regular) because mountain wake effects might play a role at those farms as well.

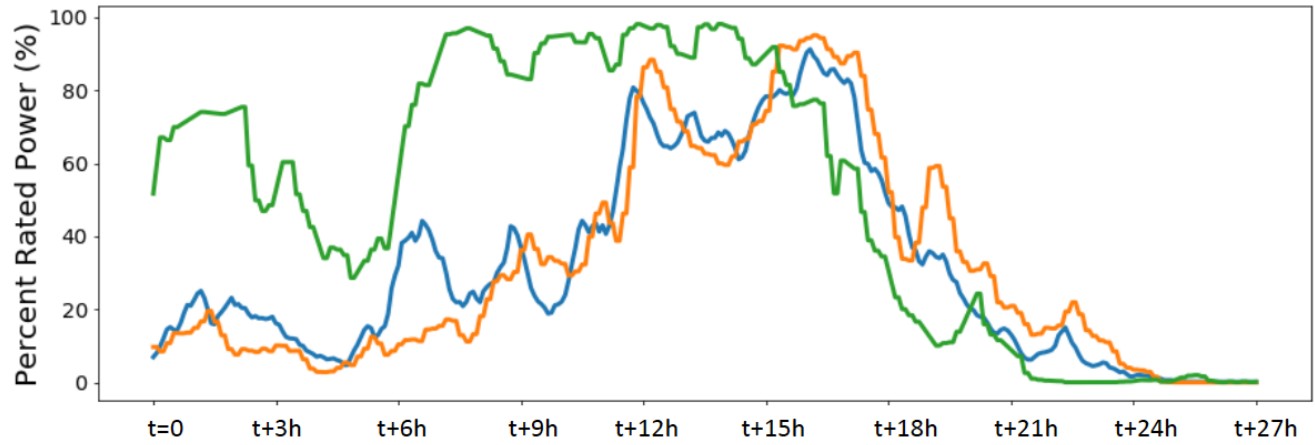


**Figure 13. Time series of total power output of the wind farm used in this study (green), and two other wind farms in the area (orange and blue). The values on the x-axis show time at 3 h intervals.**

## 4 Discussion

The previous sections discussed a mountain wave event in the Columbia Basin through simulations and observations. The
signature of these waves was apparent in nacelle wind speeds and power observations of a wind farm in the area. In this
section, we relate our findings to practical aspects in forecasting and operations.

During the event of 24 September 2016, oscillations in power caused by mountain waves are at such a critical point in the
power curve (region 2, or the "steep part") that small oscillations about the overall mean flow can make all the difference
between full rated power or approximately 1 MW of power at any given time. In this particular case, the few meters-per-
second oscillations caused by the mountain waves have dramatic effects on power production. Even after aggregating the
power output from all turbines, the power still fluctuates approximately 25 MW from mountain waves at the wind farm. For
this wind plant, this is equivalent to production from approximately 10 turbines being added or lost during the presence of the
wave's crests and troughs (assuming all turbines are on), given that one turbine could produce approximately 2.3 MW. About
11% of the total wind farm output is being influenced by the presence of mountain waves, which is considered significant
according to the empirical threshold used in the industry that more than 10% in fluctuations (or 20 MW) is significant.

Discerning signals from mountain waves from signals caused by other phenomena in the atmosphere can be challenging. For
example, mountain waves and wakes often occur concurrently, and the signals in time series of wind speed when analysing
observations at a single site, or wind turbines, can be difficult to distinguish. Mountain wakes impacting wind turbines in the
Columbia Basin are mostly created by Mt. Hood. To the north of a Mt.-Hood wake are the Gorge gap-flow westerlies and to
the south are geostrophic southwesterlies that can sometimes mix down to the southernmost and highest-elevation turbines in
the area. The instability of the mountain wake edge leads to much volatility and it is common for entire wind power plants to
go back and forth between being in the lull and being at full power. Concurrently occurring mountain wakes and waves can

lead to high-volatility periods where forecasts range from zero to full power. Power data might also include various aspects, such as curtailments and turbine wakes. During our case study, the wind farm was not curtailed. Additionally, analyses of the simulated spatial wind field, as well as cloud cover by GOES satellite data (not shown), indicate that the mountain waves appeared to shift around without systematic upstream or downstream propagation on 24 September 2016. This points to nonlinear interactions between different waves, and that the dominant dynamics are nonlinear (Nance and Durran 1997, Nance and Durran 1998, Part II). Nonetheless, mountain waves showed up in periodic signals in wind speed and power (Figs. 9–12). We analysed the wavelengths of the mountain waves; they ranged from 8–10 km in the two case studies (24 September 2016, and 11 November 2016 (Wilczak et al. 2019)) and were well captured by the numerical simulations. Future studies will include further quantification of wavelengths and whether both shorter and longer wavelengths appear simultaneously in a wind-farm region. During periods with shorter wavelengths, only parts of a wind plant will experience low wind speeds, while other parts will be exposed to stronger winds, which can result in cancelling effects so that power output is minimally affected. During these cases, accurate mountain wave and wavelength forecasts are important for wind plant operators to save on balancing costs. On the contrary, if wavelengths are long, entire regions can oscillate between near-full and near-zero power and, in particular situations, a lull over an entire region can occur, encompassing multiple wind farms. Similarly, problems arise if wind farms are spaced apart such that two wind plants happen to be in lulls while high wind speeds occur between them. Knowing when to trust model-wavelength forecasts will be the subject of future studies. In terms of speed of wave propagation, our research has shown that not all mountain waves exhibit the same speed. In the case of 24 September 2016, the simulated waves move with a speed of approximately 1.5 m s$^{-1}$ (section 3.2.3); in the case of 11 November 2016 (Wilczak et al. 2019), the simulated waves move with 2 m s$^{-1}$ (computed the same way as in section 3.2.3). This seems to contradict our findings from analysing GOES satellite data which showed no systematic propagation. Both extended wavelength and wave speed analyses are the subjects of future work.

From an operational perspective, using high-resolution forecasts that can resolve mountain waves are crucial to predict power output at a particular wind farm. The grid spacing of the simulations should be fine enough for a forecaster to recognize mountain waves. For example, even though waves are reflected in a wind field simulated on a 3 km grid, they might not be recognized as such because the waves are too wide or missing clear distinctions between high and low wind speeds that are the result of wave crests and troughs. Sharp gradients between near-zero and near-full power need to be recognizable when a forecaster looks at model simulations. Even though it is impossible to nail down the exact location of the wave crests and troughs, speed of propagation, magnitudes of wind speeds, or wavelength, the forecaster can recognize the risk for mountain waves and associated large drops or surges in power. Additionally, if high-resolution forecasts are misinterpreted (i.e., the position of wave crests and troughs are overly trusted), they have the potential to degrade a forecast. For a forecaster, it is key to be informed about the occurrence of mountain waves in order to act (e.g., assign more balancing reserves for volatility to make sure a wide range of possible production is covered). At forecasts near full power, a mountain wave event can be indicative of reductions of power. In any event, mountain wave forecasts should not be used as deterministic solutions.

## 5 Summary and Conclusions

We have shown that mountain waves can occur frequently in areas of complex terrain and can be modelled with mesoscale models as was confirmed by observations. Mountain waves can impact wind turbine and wind farm power output and, therefore, should be considered in complex terrain when designing, building, and forecasting for wind farms. Mountain waves impact the quantity of the wind resource and the quality by impacting temporal and spatial variability.

We suggest that forecasters be informed when mountain waves occur and, therefore, to be informed about wind variability to act accordingly (e.g., when setting day-ahead positions for balancing reserves and schedules). Even though the nuances of wavelength, wave propagation, or exact location are not easy to identify or simulate (because they depend on the upstream wind speed and direction as well as the vertical stability profile), being aware when mountain waves are forecast is key in operational wind energy forecasting in complex terrain. Information about the occurrence of mountain waves adds value by communicating the risk and probability of variability in power output, which helps planning for possible extreme situations. Depending on the mountain wave event and the size and shape of wind plants, effects tend to cancel out over large areas. For this to be true, wind farms should be laid out such that the windward and leeward portions are equally exposed to the mountain wave pattern. Determining the best size and orientation of wind plants to minimize mountain wave effects would be a recommendation for future studies.

Future studies should also include analyses of aggregates over larger regions to see wave patterns through wind plants as well as interactions with mountain wakes. Often, particular regions have their own peculiarities, which might also be a function of turbine age and kind, location, or elevation.

**Code/Data availability**

The measurement data that support the findings of this study are openly available in Data Archive and Portal (DAP), https://a2e.energy.gov/data. The DAP establishes a sustained data management structure with protocols and access to assure massive datasets resulting from DOE A2e (Atmosphere to Electrons) efforts will have the quality needed for scientific discovery and portals required to make data available to a broad stakeholder group. The WRF simulations and code are available from the lead author upon request. Data from wind farms are proprietary and were used under license for this study, and are therefore not publicly available.

**Author contributions**

Caroline Draxl – prepared the manuscript with contributions from all co-authors; conceptualization, formal analysis, investigation, project administration, resources, software, supervision, validation, visualization, writing (original draft, review, and editing)

Rochelle Worsnop – conceptualization, formal analysis, data curation, investigation, resources, software, validation, visualization, writing (original draft, review, and editing)

Geng Xia – formal analysis, data curation, investigation, resources, software, validation, visualization, writing (original draft)

Yelena Pichugina – formal analysis, data curation, investigation, resources, software, validation, visualization, writing (original draft)

Duli Chand – formal analysis, data curation, investigation, resources, software, visualization, writing (original draft)

Julie Lundquist – conceptualization, formal analysis, writing (original draft, review, editing)

Justin Sharp – conceptualization, writing (original draft)

Garrett Wedam – conceptualization, writing (original draft)

James Wilczak – conceptualization, formal analysis, writing (original draft, review, editing)

Larry Berg – conceptualization

The authors declare that they have no conflict of interest.

**Acknowledgements**

The authors thank the WFIP2-experiment participants who aided in the deployment and the collection of remote sensing data and our colleagues who monitored, quality controlled and provided data to the data archive portal (DAP). A special thanks go to Joe Cline (DOE), Melinda Marquis (NOAA), and Jim McCaa (Vaisala) for their effort to propose, design, and lead the WFIP2. The research was performed using computational resources sponsored by the Department of Energy's Office of Energy Efficiency and Renewable Energy and located at the National Renewable Energy Laboratory. Lastly, we thank two anonymous reviewers for their comments, which helped improve the manuscript.

This work was authored in part by the National Renewable Energy Laboratory, operated by Alliance for Sustainable Energy, LLC, for the U.S. Department of Energy (DOE) under Contract No. DE-AC36-08GO28308. Funding was provided by the U.S. Department of Energy Office of Energy Efficiency and Renewable Energy Wind Energy Technologies Office. This work was partially supported by the National Oceanic and Atmospheric Administration (NOAA) Atmospheric Science for Renewable Energy (ASRE) program. The views expressed in the article do not necessarily represent the views of the DOE or the U.S. Government. The U.S. Government retains and the publisher, by accepting the article for publication, acknowledges that the U.S. Government retains a nonexclusive, paid-up, irrevocable, worldwide license to publish or reproduce the published

form of this work, or allow others to do so, for U.S. Government purposes. A portion of the research was performed using computational resources sponsored by the Department of Energy's Office of Energy Efficiency and Renewable Energy and located at the National Renewable Energy Laboratory.

## Appendix


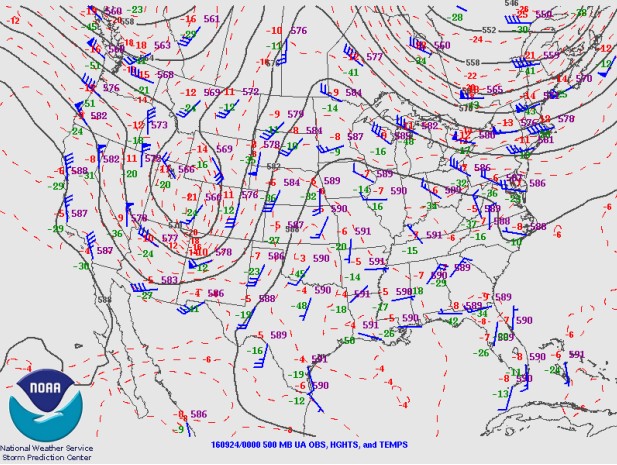

**Figure 14. 500 hPa pressure (black lines), temperature (dashed red), wind barbs (blue), temperature (red numbers), and dew point (green numbers).** *Source: National Weather Service*

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
