# Peer review of "Mountain waves can impact wind power generation"

_Wind Energy Science, 2020_

## Referee Comment (RC1) · Anonymous Referee #1 · 4 Jun 2020

Review of the manuscript WES-2020-77

Mountain waves impact wind power generation

by C. Draxl et al

Summary This paper puts together WRF model simulations and field campaign data from the WFIP2 campaign and satellite observations to illustrate the role of mountain waves in stably stratified conditions on wind oscillations in wind parks and consequences for wind energy production and its forecasting. I find the paper addresses a relevant and relatively new topic and the authors have a unique set of data (model and observations) available to study this topic. At the same time I think the paper has to become more mature before it can be published. Overall I have the feeling the model results and observations offers much more opportunity for discussion and that there is more to be learnt from the data offered.

[Figure]

Recommendations: major revisions

Major remarks:

1. Abstract should be rewritten. The sentence "This paper aims at understanding how mountain waves form in the complex terrain of the Columbia Basin, subsequently affect wind energy production, and impact aspects of operational forecasting, wind power plant layout, and integration of power into the electrical grid." is more like an announcement of the goal of the study, but in the abstract the reader expects concretely what has been learnt about from the study.

2. Title: I recommend to revise the title. The current title kind of gives the reader the expectation that a general and new theory is presented. However, the paper itself report mainly about a WRF study for a single case study for a specific region where the campaign was held. So these aspects should be included/reflected in the manuscript title. Otherwise it puts the reader on the wrong track.

3. I have the feeling that papers was put together somewhat hastily in the sense that there are many figures, while many of those figures are not discussed in depth and only touched superficially or briefly. E.g. figure 8 is discussed on only 5 lines, figure 9 in 5 lines and figure 10 in 7 lines, Figure 11 in 4 lines, fig 12 in 8 lines, Figure 13 in 5 lines ... So overall I have the impression that not all graphs are necessary or their discussions should be deepened.

4. The paper can be strengthened to include more information about the upscaling of the result. The authors elaborate on one case out of two they know when the mountain waves are relevant. It would be interesting whether the authors can say something about how often this occurs based on e.g. ERA5 data (so in a climatological sense) and how much forecast error is expected also in power forecast. Or how often the error is in a certain range (contingency table). This helps to put the paper in a broader perspective than the case study it is now.
Minor remarks:

Ln 18: Large mountains: perhaps it is good to add some scale that you mean with "large". You only mean Rockies or would the Alps or Pyrenees also be considered? Where is the cut off?

Ln 136: Mellow -> Mellor

Ln 140: please add a few sentences why Allaerts et al was so positive about this setup that you used it here as well.

Ln 169: perhaps good to add the formula for the Scorer parameter, since I can image that it is not common for the complete readership you want to serve.

Figure 6: it would be helpful to have line in panel a that shows where cross sections b) and c) are taken.

Figure 6a: it would be helpful to have the terrain height in contours plotted in this panel as well.

Figure 8: could the panels b and c be merged? Simply two lines in one panel is much more easy to compare the evolution of the profile.

Section 3.2.2: I find that section be written much more in a quantitative way, with more information about the wavelengths involved and how much WRF overestimated the wind speed in m/s at which particular height. Also how do I see the waves in these plots? I do see alternating wind speeds at some levels, but are these waves or

Ln 260: large: please add the scale you mean with large wavelengths.

Ln 268: From the spatial pattern of mountain waves in the 100 m wind speeds: why 100 m? Is the picture consistent with the behaviour at 50 and at 200 m? Also I miss a discussion here how the model resolution may have limited the wave behaviour scale. If these are horizontally propagating waves one could also use the WRF tslist tool to include multiple receptors point on the line where the FFT was performed. The

advantage of this is the higher time resolution that is obtained so one is less restricted to the model resolution.

Ln 273: "Figure 12 (a) and 12 (b) show the Hovmoller diagram of the original and reconstructed hub-height wind speed at the targeted latitude, respectively." This sentence is technically a caption and should not be part of the main text.

Ln 283: a wave period of 1.5 h: this is inconsistent with Fig 12 c where I do not see a peak in the spectrum at 1.5 h, neither in the model nor the observations.

Ln 284: The results seem to be sensitive to the chosen grid point and the period of interest (not shown). This is a critical sentence that one should elaborate on. If the result is sensitive to the location, then it is also important for the forecasting. So the reader should get a better insight here how much this sensitivity is.

Ln 366: typo in teshold

Figure 10: in panel b the horizontal white lines are dotted lines where in the others they are full lines.

Figure 11+12: s in ms-1 should not be italic along the color bar.

Figure 12: Panel c: please add the time period over which the spectrum was taken in the caption

Figure 12c: I am concerned with this plot, it is dominated by the energy at the diurnal cycle, (wave period of 24 h) but this is not the focus of the paper. As such the high energy peaks due to the wave are not easily identifiable. So I suggest to reshape this plot such that it better serves the aim of the paper.

Figure 13b: I do not see the use of this panel. Lots of info is redundant w.r.t. to panel a. If you want to show the shifted line, then just plot it in panel a, in dashed or dotted blue line.

Figure 15: The caption gives very limited information. Please add the time period or

[Figure]

model initialization time and precise location of the three sites. Otherwise this work is not reproducible.

---

## Referee Comment (RC2) · Anonymous Referee #2 · 6 Jul 2020

**1   General comments**

The paper present an interesting overview of the effect of mountain induced waves on wind farm power generation. It is an easy read and an interesting topic. I do think with some work the paper could be sharpened and for such a short paper the number of figures seems excessive. I also have some issues with the WRF model setup.

**2   Specific comments**

1. The WRF description lacks a lot of details and could be inadequate to model the phenomena described in the paper (difficult to judge due to missing description). For example, the ERA-interim data are used as boundary conditions, but these

are at a 80 km resolution so they will lack many of scales between 80 and the 3 km that is the outer domain size of the WRF simulations used in the study. The outer domain should generally be at similar spacing of the reanalysis data used for the boundary conditions and then gradually refined using nested domains. If not, a larger buffer zone needs to be present between the reanalysis to the site of interest, but this seems not to be the case judging from Fig. 3, since winds are presumable mostly from the west. There is a larger distance between the 3 km and 750 m domain at the east side, but this is not useful because the wind is not coming from that direction. It is mentioned that there is a description in Allaerts et al., but that paper is only submitted, so as long as it is not accepted it has to be presented in this paper first. Because the paper specifically deals with waves at the scales of 10-20 km, it seems that these need to be properly resolved.

2. Many of the plots are not really needed: for example, figure 7 and 8 are only discussed in a couple of lines. The message of these plots could easily be replaced by a few lines of text.

**3  Technical corrections**

- l60: There is no Wells et al. in the references

- l71: Remove space before point

- l108: Was there any filtering with respect to CNR threshold or other quality control?

- l115: Please add reference for filtering/setup of the sodar.

- l210: Brunt-Vaisala -> Brunt-Väisälä

- l132: What is temporal resolution of GOES-14?

- l136: Mellow->Mellor + add reference to the PBL scheme (Nakanishi et al.).

- l139: Isn't it usual to switch of the cumulus schemes already at those resolutions?

- l144: Was there any grid or spectral nudging performed? I think it is a good practice anyway to include the WRF namelist, because then people can easily reproduce the results.

- l156: What was their criteria for defining topographic wakes?

- l180: left) -> (Fig. 6, left)?

- l205: Fig 6. panel c: I think this would be easier to see when you plot vertical motions as red (positive) and blue (negative). Now it seems like there is only positive motions and it is not easy to distinguish wave patterns.

- l210: Brunt-Vaisala -> Brunt-Väisälä

- l281: It is not clear to me how the reconstruction is done. Please clarify.

- Fig 11b: units missing for colorbar.

- l303: Hovmoller -> Hövmöller (occurs in more places)

- l326: approximately10 -> approximately 10

- l340: Wake effects play a role at all farms, I assume. Please explain this in more detail.

---

## Author Response (AR1)

**Answer to reviewer's comments:**

*We thank the reviewers for their time and effort in reviewing this paper, for their thoughtful comments and suggestions, which have helped turn this into a higher-quality paper. Please find below detailed responses to the reviewers' comments.*

**Anonymous Referee #1**

Summary This paper puts together WRF model simulations and field campaign data from the WFIP2 campaign and satellite observations to illustrate the role of moun- tain waves in stably stratified conditions on wind oscillations in wind parks and conse- quences for wind energy production and its forecasting. I find the paper addresses a relevant and relatively new topic and the authors have a unique set of data (model and observations) available to study this topic. At the same time I think the paper has to become more mature before it can be published. Overall I have the feeling the model results and observations offers much more opportunity for discussion and that there is more to be learnt from the data offered.

Thank you for your time and efforts to review this paper. We have addressed your specific comments below.

**Major remarks:**

1. Abstract should be rewritten. The sentence "This paper aims at understanding how mountain waves form in the complex terrain of the Columbia Basin, subsequently affect wind energy production, and impact aspects of operational forecasting, wind power plant layout, and integration of power into the electrical grid." is more like an announcement of the goal of the study, but in the abstract the reader expects concretely what has been learnt about from the study.

Thank you for this comment. We agree and have revised the abstract accordingly. The abstract now reads:

*Large mountains can modify the weather downstream of the terrain. In particular, when stably stratified air ascends a mountain barrier, buoyancy perturbations develop. These perturbations can trigger mountain waves downstream of the mountains that can reach deep into the atmospheric boundary layer where wind turbines operate. Several such cases of mountain waves occurred during the Second Wind Forecast Improvement Project (WFIP2) in the Columbia Basin in the lee of the Cascade Mountains bounding the states of Washington and Oregon in the Pacific Northwest of the United States. Signals from the mountain waves appear in boundary-layer sodar and lidar observations as well as in nacelle wind speeds and power observations from wind plants. Weather Research and Forecasting model simulations also produce mountain waves and are compared to satellite observations, lidar, and sodar observations. Simulated mountain wave wavelengths and wave propagation speeds are analysed using the Fast Fourier Transform. We found that not all mountain waves exhibit the same speed and conclude that the speed of propagation, magnitudes of wind speeds, or wavelengths are important parameters for forecasters to recognize the risk for mountain waves and associated large drops or surges in power. When analysing wind farm power output and nacelle wind speeds, we found that even small oscillations in wind speed caused by mountain waves can induce oscillations between full rated power of a wind farm and half of the power output, depending on the position of the mountain wave's crests and troughs. For the wind plant analysed in this paper, 11% of the total wind farm output is influenced by the presence of mountain waves. Oscillations in measured wind speeds agree well with WRF simulations in timing and magnitude. We conclude that mountain waves can impact wind turbine and wind farm power output and, therefore, should be considered in complex terrain when designing, building, and forecasting for wind farms.*

2. Title: I recommend to revise the title. The current title kind of gives the reader the expectation that a general and new theory is presented. However, the paper itself report mainly about a WRF study for a single case study for a specific region where the campaign was held. So these aspects should be included/reflected in the manuscript title. Otherwise it puts the reader on the wrong track.

Thank you for this comment. You have a point in that we only analyzed case studies in the Columbia River Basin; however, as we point out in this paper, so far mountain waves have only anecdotally been recognized to impact wind farm production, and not quantified with regards to their impact on wind energy - we could not find a single paper that discussed this anywhere in the world. Our study shows that mountain waves impact wind power generation during two events (note that another paper was submitted analyzing another three events, and the WFIP2 event log marked a high number of mountain wave cases relevant to wind energy (see new Figure 3)). While we can't prove with our study that this is true everywhere in the world, we also don't say that "all" mountain waves impact wind power generation "all the time".

In the discussion section, we link the impact of mountain waves to operational forecasting aspects, and highlight the importance of considering mountain waves when designing and building wind farms as well. Many of these aspects are applicable to many wind farms, but probably not all. Ultimately, it might also depend on the peculiarities of the power market one is operating in.

Lastly, in a way, we see this paper as a wake-up call to alert wind farm operators that they have to be careful when the possibility of mountain waves arises. We therefore chose a very short and strong title.

In light of your comment, we revised the title to "Mountain waves can impact wind power generation".

3. I have the feeling that papers was put together somewhat hastily in the sense that there are many figures, while many of those figures are not discussed in depth and only touched superficially or briefly. E.g. figure 8 is discussed on only 5 lines, figure 9 in 5 lines and figure 10 in 7 lines, Figure 11 in 4 lines, fig 12 in 8 lines, Figure 13 in 5 lines ... So overall I have the impression that not all graphs are necessary or their discussions should be deepened.

Thank you for your comment. Reviewer 2 also commented on some figures not being necessary. To address these concerns, we have rearranged (and added more discussion) the text as well as reorganized the figures. In particular, we have removed Fig. 1, moved Fig. 5 to an appendix. Fig. 7 is discussed in 10 lines, Fig. 8 is now discussed in 2 paragraphs. Fig. 9 is discussed in one paragraph (8 lines), Fig. 10 is discussed in one paragraph (8 lines), Fig. 11 in 7 lines and referenced twice more, Fig. 13 is now discussed in one paragraph (8 lines).
We do not think that the number of lines used to describe a figure is indicative of whether a figure should be in the paper or not, though we hope the additional information we provided will aid a reader in their understanding. We feel that the remaining figures in this manuscript are needed to tell the story and solidify our points.

4. The paper can be strengthened to include more information about the upscaling of the result. The authors elaborate on one case out of two they know when the mountain waves are relevant. It would be interesting whether the authors can say something about how often this occurs based on e.g. ERA5 data (so in a climatological sense) and how much forecast error is expected also in power forecast. Or how often the error is in a certain range (contingency table). This helps to put the paper in a broader perspective than the case study it is now.

Thank you for your comment. Unfortunately, ERA5 data are too coarse (roughly 30 km grid spacing at these latitudes) to resolve mountain waves. However, Figure 4 presents the distribution of mountain waves during the WFIP2 field campaign, which lasted 18 months, thus includes all the seasons. From that figure it is clear that mountain waves occurred 17% of the time during these 18 months, which gives a nice overview of how prevalent they are. We included more language discussing which of these events were deemed having a high impact on wind energy forecasting, and revised Figure 4 to include this new information also visually.

Minor remarks:

Ln 18: Large mountains: perhaps it is good to add some scale that you mean with "large". You only mean Rockies or would the Alps or Pyrenees also be considered? Where is the cut off?

This is a good comment, however, the abstract is not a good place to cite papers that discuss this. Therefore, we left it by "large" in the abstract.  In general, the scale refers to many aspects, such as the aspect ratio of the width and height of the mountain (Bauer, M. H., G. J. Mayr, I. Vergeiner, and H. Pichler, 2000: Strongly Nonlinear Flow over and around a Three-Dimensional Mountain as a Function of the Horizontal Aspect Ratio. *J. Atmos. Sci.*, **57**, 3971–3991, https://doi.org/10.1175/1520-0469(2001)058<3971:SNFOAA>2.0.CO;2.), in combination with certain conditions of the atmosphere that we have listed by citing Reichman (1978)  and Mastaler and Renno (2003). For example, if the width of the mountain is too small, air may go around the mountain instead of over it. A good measure for this is the nondimensional mountain height, which is explained in Bauer et al. 2000. Their figure 13 gives a good visual explanation of this.
We have refrained from adding more information in the text as we have cited Reinecke and Durran (2007), who in turn have cited Bauer et al. 2000.

Ln 136: Mellow -> Mellor
Corrected as suggested

Ln 140: please add a few sentences why Allaerts et al was so positive about this setup that you used it here as well.

This setup has been used for many years in the Department of Energy Mesoscale-to-Microscale coupling project we have been participating in to conduct mesoscale simulations. It was constructed with input from researchers in the project based on their experience, and has been successfully used since. Allaers et al. is one example where it was used. We have updated the explanation in the paper and cited a report that better explains the project.

Ln 169: perhaps good to add the formula for the Scorer parameter, since I can image that it is not common for the complete readership you want to serve.
Good point, thanks. We have added the formula.

Figure 6: it would be helpful to have line in panel a that shows where cross sections b) and c) are taken.
Thank you for this comment. We have added the line and updated the figure caption.

Figure 6a: it would be helpful to have the terrain height in contours plotted in this panel as well.
Thank you for this comment. We have added terrain height and updated the caption.

Figure 8: could the panels b and c be merged? Simply two lines in one panel is much more easy to compare the evolution of the profile.
Yes, we agree, we changed the whole figure.

Section 3.2.2: I find that section be written much more in a quantitative way, with more information about the wavelengths involved and how much WRF overestimated the wind speed in m/s at which particular height. Also how do I see the waves in these plots? I do see alternating wind speeds at some levels, but are these waves or

This paragraph starts out with "Observations at fixed locations (such as from lidar or sodar) can reveal the presence of trapped lee waves through temporal fluctuations in the lee-wave pattern. Periods of alternating high and low wind speeds were observed at Wasco from all collocated remote sensing instruments as well as in the simulated horizontal wind field", which explains how you can see waves in these plots, namely through alternating high and low wind speed bands. We added some text to make it clearer how to see waves in these plots: "Good agreement is found between data from all instruments (Figure 10a–c), as waves manifest in all instruments starting near 0200 UTC, increasing in amplitude until a maximum near 1000 UTC, then decreasing." This plot is to show the signature of the waves in the observed lidar and sodar observations. In this section, we prefer to keep it qualitatively, because, as we explain in the discussion "Even though it is impossible to nail down the exact location of the wave crests and troughs, speed of propagation, magnitudes of wind speeds, or wavelength, …", the exact timing of the waves are hard to

pinpoint. This discrepancy between model and observations should be recognized and in that sense a quantitative error analysis would be beyond the scope of this paper.

Ln 260: large: please add the scale you mean with large wavelengths.

Thank you for your comment. Large wavelengths correspond to wavelengths larger than 18 km, which is the upper limit for the band pass filtering in this work. Corresponding texts have been changed in the manuscript.

Ln 268: From the spatial pattern of mountain waves in the 100 m wind speeds: why 100 m? Is the picture consistent with the behaviour at 50 and at 200 m? Also I miss a discussion here how the model resolution may have limited the wave behaviour scale. If these are horizontally propagating waves one could also use the WRF tslist tool to include multiple receptors point on the line where the FFT was performed. The advantage of this is the higher time resolution that is obtained so one is less restricted to the model resolution.

Thank you for your comment. We choose the 100m wind speed because that is the hub-height level for most of the wind turbines in the WFIP2 region. Below you can see similar plots but from 50m and 200m above ground level. As you can see, the spatial pattern of the waves are similar to that at the 100m, except that the magnitude of the wind speed increases as height increases. We added a sentence to that effect to the text.

[Figure]

The advantage of the WRF tslist tool is to get higher temporal resolution of the wind speed. However, the power variance associated with the high frequency waves (less than 1hour) is very small. Therefore, we think that the impacts of having higher time resolution will have negligible impacts on our results.

Ln 273: "Figure 12 (a) and 12 (b) show the Hovmoller diagram of the original and recon- structed hub-height wind speed at the targeted latitude, respectively." This sentence is technically a caption and should not be part of the main text.

Thank you for your comment. We understand your concern but we still think it is important to talk about the content of the figure before discussing it. We reworded the sentence to make it sound less like a caption.

*"To confirm our choice of wavelength range, we show Hovmoller diagrams of the original and reconstructed hub-height wind speed (Figure 12) at the targeted latitude."*

Ln 283: a wave period of 1.5 h: this is inconsistent with Fig 12 c where I do not see a peak in the spectrum at 1.5 h, neither in the model nor the observations.

Thank you for your comment. Yes, you are right. It should be 2.5h, not 1.5h. Below is the revised plot which has removed the 24 hour and 12 hour spikes from both the observed and simulated wind speed (following your comments on Figure 12c). Within our period of interest (1 to 4 hour), the dominant mountain wave period is 2.5 hour.

[Figure]

Ln 284: The results seem to be sensitive to the chosen grid point and the period of interest (not shown). This is a critical sentence that one should elaborate on. If the result is sensitive to the location, then it is also important for the forecasting. So the reader should get a better insight here how much this sensitivity is.

Thank you for your comment. Figure a) below shows the original plot from the manuscript while figure b) is a similar plot but the examined grid point is about 10 points (~7.5 km) away from that of Figure a). You can see that the corresponding pattern with the observations is stronger in Figure a) than Figure b), indicating the result is a bit sensitive to the chosen grid point. However, both patterns still match well with the observations. We have added a few more sentences to discuss this issue.

[Figure]

[Figure]

Ln 366: typo in teshold
Corrected

Figure 10: in panel b the horizontal white lines are dotted lines where in the others they are full lines.
We corrected the figure, thanks for pointing this out.

Figure 11+12: s in ms-1 should not be italic along the color bar.
Thank you for your comment. Corresponding changes have been made.

Figure 12: Panel c: please add the time period over which the spectrum was taken in the caption
Thank you for your comment. The revised plot has added dashed lines to indicate the period of interest.

Figure 12c: I am concerned with this plot, it is dominated by the energy at the diurnal cycle, (wave period of 24 h) but this is not the focus of the paper. As such the high energy peaks due to the wave are not easily identifiable. So I suggest to reshape this plot such that it better serves the aim of the paper.

Thank you for your comment. The lower frequency waves (24 hour and 12 hour) have been removed from the plot (Figure 12c) to better focus on those high energy peaks as suggested.

Figure 13b: I do not see the use of this panel. Lots of info is redundant w.r.t. to panel a. If you want to show the shifted line, then just plot it in panel a, in dashed or dotted blue line.

Thank you for your comment. We have merged those two plots into one plot as suggested.

Figure 15: The caption gives very limited information. Please add the time period or model initialization time and precise location of the three sites. Otherwise this work is not reproducible.

The data shown in Fig. 15 are proprietary data (recorded power output) from wind farm operators. They are not publicly available, nor can we disclose the exact location or time when the event occurred. We do believe that this plot is interesting, as it shows the impact of mountain waves on measured power output of 3 wind farms in the area. Unfortunately, most wind farm operators won't allow publishing details about their data, so the wind energy community has to find a way around displaying them by obscuring details.

**Anonymous Referee #2**

**1 General comments**

The paper present an interesting overview of the effect of mountain induced waves on wind farm power generation. It is an easy read and an interesting topic. I do think with some work the paper could be sharpened and for such a short paper the number of figures seems excessive. I also have some issues with the WRF model setup.

Thank you very much for your review, your time and efforts. We have addressed your comments below.

**2 Specific comments**

1. The WRF description lacks a lot of details and could be inadequate to model the phenomena described in the paper (difficult to judge due to missing description). For example, the ERA-interim data are used as boundary conditions, but these are at a 80 km resolution so they will lack many of scales between 80 and the 3 km that is the outer domain size of the WRF simulations used in the study. The outer domain should generally be at similar spacing of the reanalysis data used for the boundary conditions and then gradually refined using nested domains. If not, a larger buffer zone needs to be present between the reanalysis to the site of interest, but this seems not to be the case judging from Fig. 3, since winds are presumable mostly from the west. There is a larger distance between the 3 km and 750 m domain at the east side, but this is not useful because the wind is not coming from that direction. It is mentioned that there is a description in Allaerts et al., but that paper is only submitted, so as long as it is not accepted it has to be presented in this paper first. Because the paper specifically deals with waves at the scales of 10-20 km, it seems that these need to be properly resolved.

Thank you for your comment. We disagree that the outer domain should be at a similar grid spacing as the reanalysis data. This has not been followed in the literature. We have been running and designing WRF simulations for various projects without the creation of an outer nest at a similar grid spacing as the reanalysis, and even without the need of an intermediary domain. Also, in the literature you can find examples where coarse boundary conditions were downscaled without the need of an intermediary domain. As an example, Liu et al. (Liu, C., Ikeda, K., Rasmussen, R. et al. Continental-scale convection-permitting modeling of the current and future climate of North America. Clim Dyn 49, 71–95 (2017). https://doi.org/10.1007/s00382-016-3327-9) used ERA-Interim boundary conditions to run WRF on a 4-km domain. They state explicitly: "Tests showed that one-way nesting WRF, at 4-km grid spacing, with the ~75 km reanalysis was an adequate configuration without the need for a coarse grid that intermediates the ERA-Interim data and the WRF domain."
The buffer of our domain setup consists of 40/51 grid cells on the western and southern side, which is 8-10 times more than the usually recommended buffer zone of 5 grid points. The larger distance on the eastern and northern side is due to this setup being used for many studies over the WFIP2 domain. The relation between the size of inflow and outflow boundaries is not important, as long as the buffer zone from the incoming wind direction is large enough, which is the case for this study because we used 40 and 51 grid points.
In the section Code/Data Availability, we state that the namelist is available upon request. For your convenience, we pasted the relevant sections of the WRF namelist at the end of this document.

*We are also happy to announce that the paper by Allaerts et al. is now published (*http://link.springer.com/article/10.1007/s10546-020-00538-5*). However, in light of a comment of reviewer 1 we changed the reference in the paper.*

2. Many of the plots are not really needed: for example, figure 7 and 8 are only discussed in a couple of lines. The message of these plots could easily be replaced by a few lines of text.
   Reviewer 1 had a similar comment regarding too many figures. We removed Fig. 1 and moved Fig. 5 to the appendix. We added more details to the text so that Fig. 7 is discussed in 9 + 1 lines, Fig. 8 in 2 paragraphs. We have changed Fig. 8. Replacing these figures with only text would not be sufficient in our opinion as we believe they help solidify our points to the reader.

**Technical corrections**

l60: There is no Wells et al. in the references
Thanks for finding this. We have added it now.

l71: Remove space before point
 Corrected as suggested

108: Was there any filtering with respect to CNR threshold or other quality con- trol?

Basic quality control, requiring that an individual line-of-sight (LOS) velocity be measured with a carrier-to-noise ratio greater than -22 dB, has been applied to these data. The two-minute averages are based only on the 1-Hz LOS with CNR exceeding -22 dB. Lidars require a sufficient number of scatterers for a return signal, so clean air conditions have lower availability (Aitken et al. 2012). We have added this information to the text.

Aitken, M., L., M. E. Rhodes, and **J. K. Lundquist**. 2012. Performance of a wind-profiling lidar in the region of wind turbine rotor disks. Journal of Atmospheric and Oceanic Technology 29, 347-355.

115: Please add reference for filtering/setup of the sodar.
Thank you for this comment. We have added references for the sodar data.

l210: Brunt-Vaisala -> Brunt-Väisälä
Corrected

l132: What is temporal resolution of GOES-14?
*Good point, this should be mentioned in the paper. We added it (it's every 30 min).*

l136: Mellow->Mellor + add reference to the PBL scheme (Nakanishi et al.).
Corrected as suggested

l139: Isn't it usual to switch of the cumulus schemes already at those resolutions?

We believe one should have a Cumulus scheme on at dx = 3km because we are not adequately resolving many of the convective updrafts responsible for deep convection. This is why, for example, the HRRR does poorly for weakly forced air-mass thunderstorms while only getting the larger modes well (i.e., MCSs). All "mid-sized" convection in the HRRR is usually aliased up to larger scales, so it can be handled reasonably, but the scales of the updrafts are often wrong. Note that the HRRR is based on the WRF code.

Mohan and Bhati et al., (2011) analyzed the WRF model performance with parameterized cumulus in a domain over the subtropical region of Delhi, India, with a horizontal grid spacing of 2 km. They found that a combination of physical options using the Kain-Fritsch cumulus parameterization showed the best model performance during the verified period. They also found that the microphysics and cumulus parameterizations had less impact than the other physical options on the model output.

With regards to the simulated mountain wave case here, no convective precipitation is produced, so the scheme is not active. So it is practically turned off and the sentence about the cumulus scheme is not relevant. We have removed this sentence.

M. Mohan and S. Bhati, "Analysis of WRF model performance over subtropical region of Delhi, India," *Advances in Meteorology*, vol. 2011, Article ID 621235, 13 pages, 2011.

l144: Was there any grid or spectral nudging performed? I think it is a good practice anyway to include the WRF namelist, because then people can easily reproduce the results.
No, we didn't perform any nudging. As mentioned in the section about Code/Data availability, the namelist is available upon request. The relevant sections of the namelist are copied to the end of this document for your convenience.

l156: What was their criteria for defining topographic wakes?
Each week, program scientists discussed the previous 7 days, analyzing the real-time 13-km RAP, the 3-km HRRR

and 750-m HRRR-nest, global model forecasts, satellite imagery, soundings, surface observations, and wind power generation. The criteria for topographic wakes in this instance were mostly based on the available model simulations and observations: time-height plots of wind speed observations, comparison of observations in waked areas versus non-waked ones, their appearance in satellite images and the horizontal slices of simulated wind speeds (lulls). We added this information to the text.

l180: left) -> (Fig. 6, left)?
Corrected to Fig. 6 (a)

l205: Fig 6. panel c: I think this would be easier to see when you plot vertical motions as red (positive) and blue (negative). Now it seems like there is only positive motions and it is not easy to distinguish wave patterns.
Thank you for this comment. Very good point. We have adjusted the figure.

l281: It is not clear to me how the reconstruction is done. Please clarify.

Thank you for your comment. The details about the reconstruction are documented in another paper which we recently submitted to Renewable Energy (Xia et al., 2020). We have cited this work in this revision and included more details in the text.

Reference:
Xia G., Draxl C., Raghavendra A., Lundquist J.K (2020) **Validating Simulated Mountain Wave Impacts on Hub-Height Wind Speed Using SoDAR Observations,** *Submitted to Renewable Energy.*

Fig 11b: units missing for colorbar.
We added the units for the colorbar, thanks for pointing this out.

l303: Hovmoller -> Hövmöller (occurs in more places)
corrected

l326: approximately10 -> approximately 10
Corrected as suggested.

l340: Wake effects play a role at all farms, I assume. Please explain this in more detail.
Thank you for this comment. We should have been clearer here. We were talking about mountain wakes, which is now changed in the text. The exact cause of the irregularity of the other farms' oscillations would have to be looked at in more detail to draw better conclusions. However, this plot shows oscillations very nicely. As mentioned in the paper, distinguishing the effects of mountain waves, mountain wakes, and oscillations due to the meandering of the wakes is very difficult. At least for the farm discussed in detail in this paper, mountain wake effects do not play a role during the analysis time.

**WRF namelist:**

```
&domains
time_step        =  10,
max_dom          =   2,
s_we             =   1, 1,
e_we             = 381, 1001,
s_sn             =   1, 1,
e_sn             = 351, 901,
s_vert           =   1,  1,  1, 1, 1,
e_vert           =  88, 88, 88, 88, 88
eta_levels       =  1.00000, 0.99935, 0.99871, 0.99806,
                    0.99742, 0.99677, 0.99609, 0.99538,
                    0.99464, 0.99386, 0.99304, 0.99218,
                    0.99127, 0.99032, 0.98933, 0.98829,
```

```
                    0.98719,  0.98605,  0.98484,  0.98358,
                    0.98226,  0.98087,  0.97941,  0.97789,
                    0.97629,  0.97461,  0.97285,  0.97101,
                    0.96908,  0.96705,  0.96493,  0.96271,
                    0.96038,  0.95793,  0.95538,  0.95270,
                    0.94989,  0.94696,  0.94388,  0.94066,
                    0.93729,  0.93360,  0.92955,  0.92512,
                    0.92026,  0.91495,  0.90914,  0.90278,
                    0.89584,  0.88825,  0.87997,  0.87095,
                    0.86112,  0.85042,  0.83879,  0.82617,
                    0.81247,  0.79764,  0.78161,  0.76430,
                    0.74566,  0.72562,  0.70412,  0.68112,
                    0.65658,  0.63048,  0.60281,  0.57359,
                    0.54285,  0.51066,  0.47711,  0.44234,
                    0.40652,  0.36985,  0.33259,  0.29501,
                    0.25746,  0.22028,  0.18491,  0.15347,
                    0.12553,  0.10069,  0.07861,  0.05898,
                    0.04154,  0.02603,  0.01225,  0.00000,
 p_top_requested        = 10000,
 num_metgrid_levels      = 38,
 num_metgrid_soil_levels  = 4,
 dx               = 3000, 750,
 dy               = 3000, 750,
 grid_id             =   1,  2,
 parent_id            =   1,  1,
 i_parent_start        =   1, 51,
 j_parent_start        =   1, 40,
 parent_grid_ratio      =   1,  4,
 parent_time_step_ratio   =   1,  4,
 feedback            =   0,
 smooth_option         =   0,
 /
 &physics
 mp_physics           = 10, 10,
 ra_lw_physics         =  4,  4,
 ra_sw_physics         =  4,  4,
 radt              = 20, 5,
 sf_sfclay_physics       =  5,  5,
 sf_surface_physics      =  2,  2,
 bl_pbl_physics         =  5,  5,
 bldt              =  0,  0,
 cu_physics           =  1,  0,
 cudt              =  5,  0,
 isfflx             =  1,
 ifsnow             =  0,
 icloud             =  0,
 surface_input_source     =  1,
 num_soil_layers        =  4,
 sf_urban_physics        =  0,  0,
 num_land_cat          = 24,
 /
 &dynamics
 w_damping           =   1,
 diff_opt            =   1,  1,
 km_opt             =   4,  4,
 diff_6th_opt          =   2,  2,
 diff_6th_factor        = 0.12,0.12,
 base_temp           = 290.
 damp_opt            = 3,
 zdamp             = 5000.,5000.,
```

```
dampcoef              =  0.2,  0.2,
khdif                 =   0,   0,
kvdif                 =   0,   0,
non_hydrostatic       = .true.,.true.,
moist_adv_opt         =    1,    1,
scalar_adv_opt        =    1,    1,
tke_adv_opt           =    1,    1,
h_mom_adv_order       =    5,    5,
v_mom_adv_order       =    3,    3,
h_sca_adv_order       =    5,    5,
v_sca_adv_order       =    3,    3,
/
&bdy_control
spec_bdy_width        = 5,
spec_zone             = 1,
relax_zone            = 4,
specified             = .true.,.false.,
nested                = .false., .true.,
/
```

[revised manuscript text omitted]

---

## Author Response (AR2)

**Answer to reviewers' comments:**

Thank you very much for your time to review our paper again. Please find below detailed responses to your remaining comments.

Summary
The authors have substantially revised the manuscript and provided significant answers to the raised concerns by the 2 reviewers. I recommend to accept this paper for publication after minor revisions.

Remaining remarks

1. Abstract: the abstract mentions several times the wave speed. Please make clear whether you mean phase speed or group velocity.

We mean group velocities and have adjusted the text accordingly.

2. Ln 32: WRF: the acronym WRF was not introduced yet (but of course I understand what you mean).

Thanks for catching that oversight. We added the acronym on line 23.

3. My previous remark labeled as "Ln 18" about large mountains, i.e. what is meant by "large" and a recommendation to add some words about the scales you intend to discuss. The authors now refer in the response to another paper that they cited and in that one a second paper is cited and the reader has to obtain the information in this indirect way. This will not work in practise so I still recommend to add some words about this directly.

The abstract is not a good place to add references or explain scales. We agree that the word "large" is misleading. In response to your comment, we have taken out the word "large", because, in fact, mountains of any size can impact the flow. The development of mountain waves with respect to several aspects has been discussed in the introduction and in section 3.1, and several references are included.

4. Concerning the introduction of the report by Haupt et al: I agree this is a good reference, but again as a reader I do not want to be redirected to another source when I want to know the basic advantages of the set up advocated in Haupt et al. At the same time I do not ask for a complete review of WRF settings, just some justification/illustration what were the key ingredients to success in the current set up (i.e. MYNN, the height of the lowest grid point, ...). So some words like MYNN outperformed the other PBL schemes with a MAE of xxx m/s while for the other we found yyy m/s. Or MYNN was especially better for stable stratification compared to non-local schemes...

The model setup used for this study was constructed with input from modeling experts. To convey the message of this paper it is not crucial to find the best model setup to proof our point, but use one that performs well in this area and terrain. The one we have been using successfully for various projects suits this study well. We have been using the MYNN scheme consistently, as it was improved upon during the WFIP 2 project, and because, as a local scheme, allows for output of TKE. The amount of vertical levels and their spacing was necessary to resolve the turbine rotor layer and derive boundary layer quantities at relatively high vertical resolution. We have now added language regarding the use of MYNN, and restructured the sentence about the vertical resolution within the turbine rotor layer.

5. Ln 49-51: here the text mainly addresses vertically propagating waves, while the abstract mainly discusses waves that propagate downwind. This might create some confusion to get the reader on the right track.

On line 49, we have now added information that the waves can propagate downwind as well.

[revised manuscript text omitted]